# Approximations in Probabilistic Programs

Ekansh Sharma
ekansh@cs.toronto.edu
University of Toronto
Vector Institute

Daniel M. Roy
droy@utstat.toronto.edu
University of Toronto
Vector Institute

## Abstract

We introduce a new language construct, **stat**, which converts the description of the Markov kernel of an ergodic Markov chain into a sample from its unique stationary distribution. Up to minor changes in how certain error conditions are handled, we show that language constructs for soft-conditioning and normalization can be compiled away from the extended language. We then explore the problem of approximately implementing the semantics of the language with potentially nested **stat** expressions, in a language without **stat**. For a single **stat** term, the natural unrolling yields provable guarantees in an asymptotic sense. In the general case, under uniform ergodicity assumptions, we are able to give quantitative error bounds and convergence results for the approximate implementation of the extended first-order language. We leave open the question of whether the same guarantees can be made assuming mere geometric ergodicity.

## 1 Introduction

Approximations are ubiquitous for any practical implementation of a probabilistic programming language (PPL) for Bayesian modelling; This is because computing the normalized posterior of a Bayesian model is intractable. Broadly speaking, there are two types of implementations for computing the "approximate" posterior: 1) Languages like Stan, Church, and Venture use versions of Markov chain Monte Carlo (MCMC) algorithm to approximate the posterior; 2) Languages like Tensorflow Probability and Pyro use variational inference to approximate the posterior [Abadi et al. 2015; Bingham et al. 2019; Carpenter et al. 2017; Goodman et al. 2012; Tolpin et al. 2016]. A reasonable question to ask is: Can probabilistic programming systems quantify the error induced by these approximations? Also, do we know how the error scales under composition of multiple "approximate" programs and nested queries?

The answer to both the questions is no. One reason is that the semantics of probabilistic languages is not amenable to approximations introduced by the compiler of these real world languages. In this paper we bridge the gap between ideal semantics of probabilistic programming languages and approximations induced by compilers that use MCMC based inference engines. We do this by proposing a new language construct, **stat**, that takes as input an initial distribution and

a Markov kernel, and outputs the unique stationary distribution corresponding to the Markov kernel, if there exists one. We show that having **stat** in the language is "essentially" equivalent to having constructs **norm** and **score** that compute the posterior distribution. Then in Section 5.1, we give the approximate compiler for the **stat** construct based on a unrolling scheme. We then identify some semantic constraints on the Markov kernel given as argument to the **stat** construct under which we can derive quantitative error bounds on a program.

## 2 Related Work

This work builds on top of the foundations laid by Staton [2017]; Staton et al. [2016] that gives semantics to the first order probabilistic languages with construct **norm** and **score**. Previously Borgström et al. [2016]; Hur et al. [2015]; Ścibior et al. [2017] have proves the asymptotic correctness of Markov Chain Monte Carlo based inference algorithms, but the do not quantify the error due to finite computation.

Rainforth [2018] gives quantitative bounds on the error due to nested Monte Carlo approximations in probabilistic programs; But this work assumes that we can produce exact samples when the queries are nested.

In the Markov chain literature, following papers study the convergence of Markov chain when the transition kernel is approximate [Medina-Aguayo et al. 2018; Roberts et al. 1998]. This is relevant for nested queries. Also, Medina-Aguayo et al. [2018] gives quantitative convergence bounds for Metropolis–Hastings algorithm when the acceptance probability can only be accessed in an approximate manner.

## 3 Language for MCMC inference

We first give an idealized first-order probabilistic language with the proposed construct **stat** that takes as input a transition kernel for a Markov chain on some state space and returns the stationary distribution associated with the Markov chain. The language we present is based on the first-order probabilistic language introduced and studied by Staton et al. [2016] and Staton [2017], which has constructs for sampling, soft constraints, and normalization. The key differences, which we highlight again below, are (i) a syntactic distinction between probabilistic terms with and without soft constraints, which affects also typing, and (ii) the introduction of the new construct, **stat**.

*PTML'19, December 14, 2019, Vancouver, BC, Canada*
2019.

*Types:*

$$\mathbb{A}, \mathbb{A}_1 ::= \mathbb{R} \mid \mathbf{1} \mid \mathrm{P}(\mathbb{A}) \mid \mathbb{A}_0 \times \mathbb{A}_1 \mid \sum_{i \in \mathbb{N}} \mathbb{A}_i$$

*Terms:*

  *deterministic:*

$$a_0, a_1 ::= x \mid * \mid (a_0, a_1) \mid (i, a) \mid \pi_j(a) \mid f(a)$$
$$\mid \textbf{case } a \textbf{ of } \{(i, x) \Rightarrow a_i\}_{i \in I}$$

  *purely probabilistic:*

$$t_0, t_1 ::= \textbf{sample}(a) \mid \textbf{return}(a) \mid \textbf{let } x = t_0 \textbf{ in } t_1$$
$$\mid \textbf{case } a \textbf{ of } \{(i, x) \Rightarrow t_i\}_{i \in I}$$
$$\mid \textbf{stat}(t_0, \lambda x. t_1) \mid \textbf{norm}(v)$$

  *probabilistic:*

$$v_0, v_1 ::= t \mid \textbf{let } x = v_0 \textbf{ in } v_1$$
$$\mid \textbf{case } a \textbf{ of } \{(i, x) \Rightarrow v_i\}_{i \in I}$$
$$\mid \textbf{score}(a)$$

*Program:*

  $t$ is a program if $t$ is a purely probabilistic

       with no free variables

**Figure 1.** Syntax for the probabilistic language: $\mathcal{L}_{\textbf{norm}}$

### 3.1 First order language with "stat"

We begin with types and syntax of the language, presented in Figure 1. For the remainder of this paper we call this language $\mathcal{L}_{\textbf{norm}}$. Along with standard types, this language has a type $\mathbb{R}$ for real numbers and a type $\mathrm{P}(\mathbb{A})$ as a type for the space of probability measures on $\mathbb{A}$. The language has all the basic constructors, destructor, case statements, and sequencing. Along with standard programming language constructs, the language has probabilistic features including **sample** statements that takes in a probability measure as input and returns a sample from it, **score** statements that scales the prior program with likelihood, and **norm** term that takes an un-normalized measure and returns the normalized probability measure.

Here, we give a very brief review of the semantics of the language. For a detailed account, we refer the readers to Staton [2017]. Types in the language are interpreted as measurable spaces $(\llbracket \mathbb{A} \rrbracket, \Sigma_{\llbracket \mathbb{A} \rrbracket})$. As in [Staton et al. 2016], each term in the language is either *deterministic* or *probabilistic*, satisfying typing judgments of the form $\Gamma \vdash_d t : \mathbb{A}$ and $\Gamma \vdash_p t : \mathbb{A}$, respectively, given some environment/context $\Gamma = (x_1 : \mathbb{A}_1, ..., x_n : \mathbb{A}_n)$. Letting $\llbracket \Gamma \rrbracket = \prod_{i=1}^n \llbracket \mathbb{A}_i \rrbracket$, a deterministic term denotes a measurable function from the environment $\llbracket \Gamma \rrbracket$ to $\llbracket \mathbb{A} \rrbracket$. As in [Staton 2017], a probabilistic term denotes an $S$-finite kernel from $\llbracket \Gamma \rrbracket$ to $\llbracket \mathbb{A} \rrbracket$. Different

from Staton [2017]; Staton et al. [2016], we distinguish a subset of probabilistic terms we call *purely probabilistic*, which satisfy an additional typing judgment $\Gamma \vdash_{p1} t : \mathbb{A}$. A purely probabilistic term denotes a probability kernel from $\llbracket \Gamma \rrbracket$ to $\llbracket \mathbb{A} \rrbracket$. Departing again from Staton [2017]; Staton et al. [2016], a *program* in our language is a purely probabilistic term with no free variables.

#### 3.1.1 Sequencing and sampling terms

In addition to standard **let** statements and **return** statements for sequencing, the language has a construct for producing a random sample from a probability distribution.

As in [Staton 2017], the semantics of the **let** construct is defined in terms of integration as follows:

$$\llbracket \textbf{let } x = t_1 \textbf{ in } t_2 \rrbracket_{\gamma, A} \stackrel{\text{def}}{=} \int_{\llbracket \mathbb{A} \rrbracket} \llbracket t_2 \rrbracket_{\gamma, x, A} \llbracket t_1 \rrbracket_{\gamma, dx}$$

Since both $t_1$ and $t_2$ are probabilistic terms, both are interpreted as S-finite kernels; The category of S-finite kernels is closed under composition, thus the term **let** $x = t_1$ **in** $t_2$ is also interpreted as an S-finite kernel.

The semantics of the **return** statement is given by the kernel $\llbracket \textbf{return}(t) \rrbracket : \llbracket \Gamma \rrbracket \times \Sigma_{\llbracket \mathbb{A} \rrbracket} \to [0, 1]$

$$\llbracket \textbf{return}(t) \rrbracket_{\gamma, A} \stackrel{\text{def}}{=} \begin{cases} 1 & \text{if } \llbracket t \rrbracket_\gamma \in A \\ 0 & \text{otherwise} \end{cases}$$

Finally the **sample** statement takes in as argument a deterministic term of type $\mathrm{P}(\mathbb{A})$ that is it takes in as argument a probability measure on the space $\llbracket \mathbb{A} \rrbracket$. Thus the semantics is given as:

$$\llbracket \textbf{sample}(t) \rrbracket_{\gamma, A} = \llbracket t \rrbracket_{\gamma, A},$$

where $\llbracket t \rrbracket_\gamma \in \llbracket \mathrm{P}(\mathbb{A}) \rrbracket$.

#### 3.1.2 Soft constraints and normalization terms

We are studying a probabilistic language for Bayesian inference; We have terms in the language that is used to scale the prior by the likelihood of some observed data and a term that re-normalizes the scaled measure to return the posterior distribution over the return type. The constructs **score** and **norm** are the constructs that, respectively, scale the prior program, and normalize the program to return the posterior probability distribution on the output type, if there exists one. The semantics for the **score** construct are given by a S-finite kernel on the unit type, $\mathbf{1}$, as follows:

$$\llbracket \textbf{score}(a) \rrbracket_{\gamma, A} = \begin{cases} \left| \llbracket t \rrbracket_\gamma \right| & \text{if } A = \{()\} \\ 0 & \text{otherwise} \end{cases}$$

The main difference between this semantics and the denotational semantics of the language proposed in Staton [2017]; Staton et al. [2016] is in the semantics of **norm**. We interpret the semantics of **norm** terms as a probability kernel on the

sum space given as $[\![\mathbf{norm}(t)]\!] : [\![\Gamma]\!] \times \Sigma_{\mathbb{A}+1} \to [0,1]$ defined as

$$[\![\mathbf{norm}(t)]\!]_{\gamma,A} = \begin{cases} \frac{[\![t]\!]_{\gamma,\{u|(0,u)\in A\}}}{[\![t]\!]_{\gamma,[\![\mathbb{A}]\!]}} & \text{if } [\![t]\!]_{\gamma,[\![\mathbb{A}]\!]} \in (0,\infty) \\ 0 & \text{else if } (1,()) \notin A \\ 1 & \text{else if } (1,()) \in A \end{cases} ,$$

where $A \in [\![\mathbb{A}+1]\!]$. The key distinction is that we are not able to determine if the term $[\![t]\!]_\gamma$ is an infinite measure or a null measure.

### 3.1.3 Stationary terms

One of the main contributions of this paper is that we propose a new feature in the probabilistic language that takes as argument a Markov chain transition kernel on some state space and returns the stationary distribution associated with the kernel.

We do this by allowing the users to define a transition kernel on some measurable space using a standard lambda expression. Following is the syntax and typing rules for the stationary term:

$$\frac{\Gamma \Big|_{p1} t_0 : \mathbb{A} \quad \Gamma, x : \mathbb{A} \Big|_{p1} t_1 : \mathbb{A}}{\Gamma \Big|_{p1} \mathbf{stat}(t_0, \lambda x.t_1) : \mathbb{A}+1} .$$

To give the denotational semantics for the **stat**-term, we first introduce to the meta language to define the function $\mathrm{ST} : [\![P(\mathbb{A})]\!]^{[\![\mathbb{A}]\!]} \to [\![P(\mathbb{A})]\!] + \mathbf{1}$ as follows: For some $k : X \times \Sigma_X \to [0,1]$,

$$\mathrm{ST}(k) = \begin{cases} (0,\mu) & \text{if } \mu \text{ is unique and} \\ & \left\|\mu(\cdot) - \int_X k(x,\cdot)\mu(dx)\right\|_{\mathrm{tv}} = 0 \\ (1,()) & \text{otherwise} \end{cases} .$$

Now, we define the semantics to the stationary term as:

$$[\![\mathbf{stat}(t_0,\lambda x.t_1)]\!]_{\gamma,A} = \begin{cases} \mu_\gamma(A) \text{ if } \mathrm{ST}([\![t_1]\!]_\gamma) = (0,\mu_\gamma) \\ 0 & \text{if } \mathrm{ST}([\![t_1]\!]_\gamma) = (1,()) \text{ and} \\ & (1,()) \notin A \\ 1 & \text{if } \mathrm{ST}([\![t_1]\!]_\gamma) = (1,()) \text{ and} \\ & (1,()) \in A. \end{cases}$$

## 4 Removing soft constraints and normalization terms from $\mathcal{L}_{\mathrm{norm}}$

So far we have modified the probabilistic language proposed by Staton et al. [2016] to include a new construct **stat**. Consider the language in Figure 2. We call this language $\mathcal{L}_{\mathrm{stat}}$. The key difference between the $\mathcal{L}_{\mathrm{stat}}$ and $\mathcal{L}_{\mathrm{norm}}$ is that $\mathcal{L}_{\mathrm{stat}}$ does not have the language constructs for soft constraints and normalization. Following theorem says that $\mathcal{L}_{\mathrm{stat}}$ is equivalent to $\mathcal{L}_{\mathrm{norm}}$.

**Theorem 4.1** (Equivalence). *For the languages $\mathcal{L}_{\mathrm{norm}}$ and $\mathcal{L}_{\mathrm{stat}}$, there exists a function $\phi$ that maps $\mathcal{L}_{\mathrm{norm}}$-phrases to*

*Types:*

$$\mathbb{A}_0, \mathbb{A}_1 ::= \mathbb{R} \mid \mathbf{1} \mid P(\mathbb{A}) \mid \mathbb{A}_0 \times \mathbb{A}_1 \mid \sum_{i \in \mathbb{N}} \mathbb{A}_i$$

*Terms:*

  *deterministic:*

$$a_0, a_1 ::= x \mid * \mid (a_0, a_1) \mid (i, a) \mid \pi_j(a) \mid f(a)$$
$$\mid \mathbf{case}\ a\ \mathbf{of}\ \{(i, x) \Rightarrow a_i\}_{i \in I}$$

  *purely probabilistic:*

$$t_0, t_1 ::= \mathbf{sample}(a) \mid \mathbf{return}(a) \mid \mathbf{let}\ x = t_0\ \mathbf{in}\ t_1$$
$$\mid \mathbf{case}\ a\ \mathbf{of}\ \{(i, x) \Rightarrow t_i\}_{i \in I}$$
$$\mid \mathbf{stat}(t_0, \lambda x.t_1)$$

*Program:*

  *t is a program if t is a purely probabilistic*
        *with no free variables*

**Figure 2.** Syntax for the probabilistic language: $\mathcal{L}_{\mathrm{stat}}$

$\mathcal{L}_{\mathrm{stat}}$-*phrases such that under the semantics described in Section 3 following statements hold:*

- *$\phi(t)$ is an $\mathcal{L}_{\mathrm{stat}}$-program for all $\mathcal{L}_{\mathrm{norm}}$ programs $t$;*
- *$\varphi(\mathbb{F}(e_1,\ldots,e_a)) = \mathbb{F}(\varphi(e_1),\ldots,\varphi(e_a))$ for all features of $\mathcal{L}_{\mathrm{stat}}$*
- *For all programs $t$, and for all $\gamma$, $\left\|[\![t]\!]_\gamma - [\![\phi(t)]\!]_\gamma\right\|_{\mathrm{tv}} = 0$*

*Proof idea.* First, we identify that all **score** terms in a program in $\mathcal{L}_{\mathrm{norm}}$ is encapsulated in the **norm** statement. For every **norm** statement in a program, we can construct a Markov kernel that proposes a move with the prior probability distribution described by the program; Then accept/reject the move using the standard Metropolis–Hastings acceptance probability. This transition kernel has the stationary distribution that is semantically equivalent to the normalize term. Thus we can use the **stat** construct to compile away **norm** and **score**. □

## 5 Approximate compilation of probabilistic programs

We saw in Section 3, to compute the the normalized measure one term $\mathbf{norm}(t)$ from an un-normalized probabilistic term $t$, we need to compute a normalization factor $\int_{[\![\mathbb{A}]\!]} [\![t]\!]_{\gamma, dx}$. In the previous section we showed that the language $\mathcal{L}_{\mathrm{norm}}$, that includes **norm** construct, is equivalent to the language $\mathcal{L}_{\mathrm{stat}}$ that doesn't have the **norm** construct. Unfortunately, $\mathcal{L}_{\mathrm{stat}}$ still includes the language construct **stat** and computing the stationary distribution for arbitrary Markov kernels is also computationally intractable. One advantage of using **stat** construct over **norm** is that there exists approximate compilation for the **stat** construct that, under some suitable assumptions are amenable to error quantification. In this

section we begin by giving a broad semantic constraints that needs to be imposed on the Markov kernels programming language under which we can give an approximate compilation scheme for a single **stat** construct which is asymptotically exact. We later impose stricter semantic restrictions that allows us to give quantitative bounds on error bounds associated to the approximate compilation of program with multiple **stat** terms, including nested queries.

### 5.1 Approximate implementation for stat terms

Before giving an approximate compilation for a **stat** term, we first make an assumption on the inputs to the **stat** terms.

**Assumption 1.** *For a term* $\Gamma \mid_{\overline{p1}} \mathbf{stat}(t_0, \lambda x.t_1) : \mathbb{A} + 1$ *in the language, we make the following assumptions for all* $\gamma$, $[\![t_1]\!]_\gamma$ *is an ergodic Markov kernel with stationary distribution* $\pi_\gamma \in \mathcal{M}[\![\mathbb{A}]\!]$ *and* $[\![t_0]\!]_\gamma$ *is such that:*

$$\lim_{N \to \infty} \left\| \int_{[\![\mathbb{A}]\!]} [\![t_1]\!]_{\gamma,x}^N(\cdot) [\![t_0]\!]_{\gamma,dx} - \pi_\gamma(\cdot) \right\|_{\mathrm{tv}} = 0$$

Under Assumption 1, there exists a simple program transformation via iteration that approximates the **stat**-term. First, let's inductively define syntactic sugar **iterate** as follows:

$$\mathbf{iterate}^0(t_0, \lambda x.t_1) := t_0$$

$$\mathbf{iterate}^N(t_0, \lambda x.t_1) := \mathbf{let}\ x = \mathbf{iterate}^{N-1}(t_0, \lambda x.t_1)\ \mathbf{in}\ t_1$$

The semantics of the approximate program transformation is given as:

$$[\![\mathbf{iterate}^N(t_0, \lambda x.t_1)]\!]_\gamma = \int [\![t_1]\!]_{\gamma,x}^N [\![t_0]\!]_\gamma(dx)$$

Given these syntactic sugar, now we given the program transformation $\phi$ as follows:

$$\phi(\mathbf{stat}(t_0, \lambda x.t_1), n) \stackrel{\mathrm{def}}{=} \mathbf{iterate}^n(t_0, \lambda x.t_1)$$

Even though the Assumption 1 yields an asymptotically exact compilation scheme for the **stat** construct, a program with nested **stat** terms down not behave well in general. This is highlighted in the following problem.

**Problem 1.** *Semantics of the* **stat** *construct is not continuous, i.e., for some term* $\Gamma \mid_{\overline{p1}} \mathbf{stat}(t_0, \lambda x.t_1) : \mathbb{A} + 1$ *if we know that* $[\![t_1]\!]_{\gamma,x}$ *is an ergodic kernel that has a unique stationary distribution, it is possible to construct an approximate implementation of the Markov transition kernel* $\lambda x.t_1'$ *such that*

$$\exists \delta \in (0,1) \forall \gamma, x.\ \left\| [\![t_1]\!]_{\gamma,x} - [\![t_1']\!]_{\gamma,x} \right\| \leq \delta,$$

*but the Markov kernel* $[\![t_1']\!]_{\gamma,x}$ *does not have a stationary distribution. Such an example is given in proposition 1 of Roberts et al. [1998].*

To side step the issue stated in Problem 1, we need to make further semantic restrictions on the Markov kernels passed in as input to the **stat** construct. In this paper we identify that if the transition kernel given to the **stat** construct is uniformly ergodic, then **stat** construct is continuous.

**Definition 5.1.** *A Markov chain with transition kernel* $P$ : $[\![\mathbb{A}]\!] \times \Sigma_{[\![\mathbb{A}]\!]} \to [0,1]$ *is uniformly ergodic with stationary distribution* $\pi$ *if there exists* $C \in [0, \infty)$, $\rho \in [0,1)$ *such that for all* $x \in [\![\mathbb{A}]\!]$

$$\left\| P^N(x, \cdot) - \pi(\cdot) \right\|_{\mathrm{tv}} \leq C\rho^N$$

### 5.2 Quantitative error bounds

We finally state the quantitative error bounds for an approximate probabilistic programs where each **stat** term is uniformly ergodic.

**Theorem 5.2** (Quantitative error bound for probabilistic programs). *Let* $P$ *be a probabilistic program in the proposed language. Let* $\{\mathbf{stat}(t_{0i}, \lambda x.t_{1i})\}_{i \in I}$ *be the set of all stationary terms in the program* $\varnothing \mid_{\overline{p1}} P : \mathbb{B}$ *such that* $\forall \gamma$, *there exists constants* $\{C_i\}$ *and* $\{\rho_i\}$ *such that* $[\![t_{1i}]\!]_\gamma$ *is uniformly ergodic with those constants. Let* $P'$ *be a program where* $\forall i \in I$, $\mathbf{stat}(t_{0i}, \lambda x.t_{1i})$ *is replaced by* $\phi(\mathbf{stat}(t_{0i}, \lambda x.t_{1i}), N_i)$ *where* $N_i \in \mathbb{N}$, *then there exists constants* $\{C_i'\}_{i \in I}$ *such that*

$$\left\| [\![P]\!]_\gamma - [\![P']\!]_\gamma \right\|_{\mathrm{tv}} \leq \sum_{i \in I} C_i' \rho_i^{N_i}.$$

*Proof.* In appendix.                                            □

## 6 Summary and Discussion

Markov chain Monte Carlo algorithms are workhorses for approximate computation of the normalized posterior distribution. MCMC algorithms are popular because we they give us asymptotic convergence guarantees and are commonly used as "approximate" compilers for probabilistic programming languages. In this abstract we proposed a novel language construct **stat** that allows us give a formal description for such compilers. We then gave a simple compiler description that again at its core implements an MCMC algorithm. Typically quantifying the rate at which Markov chain with some given transition kernel converges is an open problem and the one we do not attempt to solve in this paper. We make a semantic assumption that the user using our language provides us with a description of the Markov kernel that converges uniformly to the corresponding target distribution. We show that under this uniform convergence property, we can derive rates at which the approximate compiler converges to the original program.

The assumption for uniform ergodicity is crucial for us to derive the quantitative bound. The main difficulty we found in relaxing the uniform ergodicity assumption is the fact that our language allows us to nest the **stat**. We leave as open problem if we can relax the uniform ergodicity assumption.

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

## A  Proof of Theorem 5.2

Before we prove the Theorem 5.2, we prove the following proposition that characterizes the uniform continuity of **let** and **case** statements under uniform ergodicity assump-tion.

**Proposition A.1.** *The following statements hold:*

1. *Let* $\left[\Gamma \mid_{\overline{p1}} t_0 : \mathbb{A}\right]$, $\left[\Gamma \mid_{\overline{p1}} t_0' : \mathbb{A}\right]$, $\left[\Gamma, x : \mathbb{A} \mid_{\overline{p1}} t_1 : \mathbb{B}\right]$, *and* $\left[\Gamma, x : \mathbb{A} \mid_{\overline{p1}} t_1' : \mathbb{B}\right]$ *be purely probabilistic terms. If for all* $\gamma \in \Gamma$, $\left\|\llbracket t_0' \rrbracket_\gamma - \llbracket t_0 \rrbracket_\gamma\right\|_{\mathrm{tv}} \leq \alpha$ *and for all*

$\llbracket x \rrbracket_\gamma \in \llbracket \mathbb{A} \rrbracket$,

$$\left\|\llbracket t_1' \rrbracket_\gamma(\llbracket x \rrbracket_\gamma) - \llbracket t_1 \rrbracket_\gamma(\llbracket x \rrbracket_\gamma)\right\|_{\mathrm{tv}} \leq \beta$$

*then*

$$\left\|\llbracket \textbf{let } x = t_0 \textbf{ in } t_1 \rrbracket_\gamma - \llbracket \textbf{let } x = t_0' \textbf{ in } t_1' \rrbracket_\gamma\right\|_{\mathrm{tv}} \leq \alpha + \beta$$

2. *Let* $\left[\Gamma, x : \mathbb{A}_i \mid_{\overline{p1}} t_i : \mathbb{B}\right]$ *and* $\left[\Gamma, x : \mathbb{A}_i \mid_{\overline{p1}} t_i' : \mathbb{B}\right]$, *such that*

$$\forall i \in I, \forall x \in \llbracket \mathbb{A}_i \rrbracket, \left\|\llbracket t_i' \rrbracket_{\gamma, x} - \llbracket t_i \rrbracket_{\gamma, x}\right\|_{\mathrm{tv}} \leq \alpha_i$$

$$\left\|\begin{array}{l}\llbracket \textbf{case } a \textbf{ in } \left\{(i, x) \Rightarrow t_i'\right\}_{i \in I}\rrbracket_\gamma - \\ \llbracket \textbf{case } a \textbf{ in } \left\{(i, x) \Rightarrow t_i\right\}_{i \in I}\rrbracket_\gamma\end{array}\right\|_{\mathrm{tv}} \leq \sup\left\{\alpha_i\right\}_{i \in I}$$

*Proof.*  1. For the **let** construct:

$$\left\|\int \llbracket t_1 \rrbracket_{\gamma, x} \llbracket t_0 \rrbracket_{\gamma, dx} - \int \llbracket t_1' \rrbracket_{\gamma, x} \llbracket t_0' \rrbracket_{\gamma, dx}\right\|_{\mathrm{tv}}$$

$$\leq \left\|\int \llbracket t_1 \rrbracket_{\gamma, x} \llbracket t_0 \rrbracket_{\gamma, dx} - \int \llbracket t_1' \rrbracket_{\gamma, x} \llbracket t_0 \rrbracket_{\gamma, dx}\right\|_{\mathrm{tv}}$$

$$+ \left\|\int \llbracket t_1' \rrbracket_{\gamma, x} \llbracket t_0 \rrbracket_{\gamma, dx} - \int \llbracket t_1' \rrbracket_{\gamma, x} \llbracket t_0' \rrbracket_{\gamma, dx}\right\|_{\mathrm{tv}}$$

$$= \left\|\int \left(\llbracket t_1 \rrbracket_{\gamma, x} - \llbracket t_1' \rrbracket_{\gamma, x}\right) \llbracket t_0 \rrbracket_{\gamma, dx}\right\|_{\mathrm{tv}}$$

$$+ \sup_A \left|\int \llbracket t_1' \rrbracket_\gamma(x, A)\llbracket t_0 \rrbracket_{\gamma, dx} - \int \llbracket t_1' \rrbracket_\gamma(x, A)\llbracket t_0' \rrbracket_{\gamma, dx}\right|$$

$$\leq \int \sup_{x'} \left\|\llbracket t_1 \rrbracket_\gamma(x') - \llbracket t_1' \rrbracket_\gamma(x')\right\|_{\mathrm{tv}} \llbracket t_0 \rrbracket_{\gamma, dx}$$

$$+ \sup_{f \leq 1} \left|\int f(x)\llbracket t_0 \rrbracket_{\gamma, dx} - \int f(x)\llbracket t_0' \rrbracket_{\gamma, dx}\right|$$

$$= \int \beta\llbracket t_0 \rrbracket_{\gamma, dx} + \alpha$$

$$= \alpha + \beta$$

2. For the **case** construct:

$$\left\|\begin{array}{l}\llbracket \textbf{case } a \textbf{ in } \left\{(i, x) \Rightarrow t_i'\right\}_{i \in I}\rrbracket_\gamma - \\ \llbracket \textbf{case } a \textbf{ in } \left\{(i, x) \Rightarrow t_i\right\}_{i \in I}\rrbracket_\gamma\end{array}\right\|_{\mathrm{tv}}$$

$$= \left\|\llbracket t_i' \rrbracket_{v, \gamma} - \llbracket t_i \rrbracket_{v, \gamma}\right\|_{\mathrm{tv}} \qquad \text{if } (i, v) = \llbracket a \rrbracket_\gamma$$

$$\leq \alpha_i \qquad \text{if } (i, v) = \llbracket a \rrbracket_\gamma$$

$$\leq \sup_i \left\{\alpha_i\right\}_{i \in I}$$

$\square$

**Remark A.2.** *A natural consequence of the sugar is that the following terms are syntactically equivalent.*

$$\textbf{iterate}^N(t_0, \lambda x.t_1) := \textbf{iterate}^{N-i}(\textbf{iterate}^i(t_0, \lambda x.t_1), \lambda x.t_1)$$

We characterize the error of an approximate **iterate** trans-formation.

**Theorem A.3.** *Let* $\Gamma, x \mid_{\overline{p1}} t_1$ *and* $\Gamma, x \mid_{\overline{p1}} t_1'$ *be probabilistic terms. If* $[\![t_1]\!]_\gamma$ *is uniformly ergodic with stationary distribution* $\pi$ *and constants* $C$, *and* $\rho$ *and*

$$\left\| [\![t_1']\!]_\gamma - [\![t_1]\!]_\gamma \right\|_{\mathrm{tv}} \le \varepsilon,$$

*then*

$$\left\| [\![\mathbf{iterate}^N(t_0, \lambda x.t_1')]\!] - [\![\mathbf{stat}(t_0, \lambda x.t_1)]\!] \right\|_{\mathrm{tv}} \le \frac{\varepsilon C}{1 - \rho} + C\rho^N.$$

We begin the proof of theorem Theorem A.3 by giving a simple contraction lemma that quantifies the error associated with the $N$ step iteration transformation with a different initial distribution.

**Lemma A.4** (Contraction). *If* $R\left([\![t_1]\!]_\gamma, \mathrm{ST}([\![t_1]\!]_\gamma), C, \rho\right)$ *then for all*

$$\left\| \begin{array}{c} [\![\mathbf{iterate}^N(m_1, \lambda x.t_1)]\!]_\gamma - \\ [\![\mathbf{iterate}^N(m_2, \lambda x.t_1)]\!]_\gamma \end{array} \right\|_{\mathrm{tv}} \le C\rho^N \left\| [\![m_1]\!]_\gamma - [\![m_2]\!]_\gamma \right\|.$$

*Proof.* The proof of this lemma follows directly from linearity of integration. □

To prove Theorem A.3, we now give following theorem quantifies the distance between iteration transformation the transition kernels are close.

**Lemma A.5.** *Let* $\Gamma, x \mid_{\overline{p1}} t_1$ *and* $\Gamma, x \mid_{\overline{p1}} t_1'$ *be probabilistic terms. If there exists* $\pi \in \mathcal{M}(\mathbb{A})$, $C \in \mathbb{R}_+$, *and* $\rho \in [0, 1)$ *such that* $R([\![t_1]\!]_\gamma, \pi, C, \rho)$ *and*

$$\left\| [\![t_1']\!]_\gamma - [\![t_1]\!]_\gamma \right\|_{\mathrm{tv}} \le \varepsilon$$

*then,*

$$\left\| [\![\mathbf{iterate}^N(t_0, \lambda x.t_1')]\!] - [\![\mathbf{iterate}^N(t_0, \lambda x.t_1)]\!] \right\|_{\mathrm{tv}} \le \frac{\varepsilon C}{1 - \rho},$$

*Proof.* We show this by first noting that

$$\left\| [\![\mathbf{iterate}^N(t_0, \lambda x.t_1))]\!]_\gamma - [\![\mathbf{iterate}^N(t_0, \lambda x.t_1')]\!]_\gamma \right\|_{\mathrm{tv}}$$

$$= \left\| \begin{array}{c} \sum_{i=0}^{N-1} [\![\mathbf{iterate}^{N-i}(\mathbf{iterate}^i(t_0, \lambda x.t_1'), \lambda x.t_1)]\!]_\gamma \\ - [\![\mathbf{iterate}^{N-i-1}(\mathbf{iterate}^{i+1}(t_0, \lambda x.t_1'), \lambda x.t_1)]\!]_\gamma \end{array} \right\|_{\mathrm{tv}}$$

$$= \left\| \begin{array}{c} \sum_{i=0}^{N-1} [\![\mathbf{iterate}^{N-i-1}(\mathbf{let}\ x = \mathbf{iterate}^i(t_0, \lambda x.t_1')\ \mathbf{in}\ n, \lambda x.t_1))]\!] \\ - [\![\mathbf{iterate}^{N-i-1}(\mathbf{iterate}^{i+1}(t_0, \lambda x.t_1'), \lambda x.t_1)]\!]_\gamma \end{array} \right\|_{\mathrm{tv}}$$

Applying the contraction lemma

$$\le \sum_{i=0}^{N-1} C\rho^{N-i-1} \left( \left\| [\![\mathbf{let}\ x = \mathbf{iterate}^i(t_0, \lambda x.t_1')\ \mathbf{in}\ n]\!]_\gamma \right. \right.$$

$$\left. \left. - [\![\mathbf{let}\ x = \mathbf{iterate}^i(t_0, \lambda x.t_1')\ \mathbf{in}\ n']\!]_\gamma \right\|_{\mathrm{tv}} \right)$$

$$\le \sum_{i=0}^{N-1} C\rho^i \varepsilon \le \frac{\varepsilon C}{1 - \rho}.$$

□

*Proof of Theorem A.3.* Given the Lemma A.5 and the assumption in the theorem statement that $R([\![t_1]\!]_\gamma, \pi, C, \rho)$ holds, the proof of theorem follows by triangle inequality. □

Now we are in a position to prove the main theorem of this section.

*Proof of Theorem 5.2.* The proof is seen by induction on probabilistic terms.

- *Base case:*

  Leaf node for the induction is the terms of the form $\mathbf{stat}(t_0', \lambda x.t_1')$ . By the assumption of uniform ergodicity there exists $C', \rho'$ such that following holds

  $$[\![\mathbf{stat}(t_0, \lambda x.t_1)]\!] - [\![\mathbf{iterate}^{N'}(t_0, \lambda x.t_1)]\!] \le C'\rho'^{N'}.$$

  Thus the base case holds.

- *Inductive Step:*

  For the inductive step we show the hypothesis holds for for all constructor of probabilistic terms in our language.

  – **case** terms: By the inductive hypothesis,

  $$\left\| [\![t_j']\!]_\gamma - [\![t_j']\!]_\gamma \right\|_{\mathrm{tv}} \le \sum_{i \in I_j} C_i \rho^{N_i}.$$

  Now, we show

  $$\left\| \begin{array}{c} [\![\mathbf{case}\ a\ \mathbf{in}\ \left\{(j, x) \Rightarrow t_j'\right\}_{j \in J}]\!] - \\ [\![\mathbf{case}\ a\ \mathbf{in}\ \left\{(j, x) \Rightarrow t_j\right\}_{j \in J}]\!] \end{array} \right\|_{\mathrm{tv}} \le \sum_{i \in \bigcup_{j \in J} I_j} C_i' \rho_i^{N_i}$$

  From Proposition A.1 and the inductive hypothesis, we know

  $$\left\| \begin{array}{c} [\![\mathbf{case}\ a\ \mathbf{in}\ \left\{(j, x) \Rightarrow t_j'\right\}_{j \in J}]\!] - \\ - [\![\mathbf{case}\ a\ \mathbf{in}\ \left\{(j, x) \Rightarrow t_j\right\}_{j \in J}]\!] \end{array} \right\|_{\mathrm{tv}} \le \sup_{j \in J} \sum_{i \in I_j} C_i \rho^{N_i}$$

  $$\le \sum_{i \in \bigcup_j I_j} \rho^{N_i}.$$

  – **let** term: We need to show

  $$\left\| [\![\mathbf{let}\ x = t_1\ \mathbf{in}\ t_2]\!] - [\![\mathbf{let}\ x = t_1'\ \mathbf{in}\ t_2']\!] \right\|_{\mathrm{tv}} \le \sum_{i \in I_1 \cup I_2} C_i \rho_i^{N_i}$$

  This follows from inductive hypothesis and Proposition A.1.

  – **stat** term: From Theorem A.3 and inductive hypothesis it follows that

  $$\left\| \begin{array}{c} [\![\mathbf{stat}(t_0, \lambda x.t_1)]\!] - \\ [\![\mathbf{iterate}^{N'}(t_0, \lambda x.t_1')]\!] \end{array} \right\|_{\mathrm{tv}} \le C'\rho'^{N_i} + \sum_i \frac{C'C_i}{1 - \rho'} \rho_i^{N_i}.$$

  □

