# OpenReview forum: "Approximations in Probabilistic Programs"
_NeurIPS.cc/2019/Workshop/Program_Transformations — Program Transformations @NeurIPS2019 Poster_

### Official Review · AnonReviewer1 · 2019-09-28
**Theoretical contribution in PPLs; well-organized, within scope, rigorous**

**Confidence:** 2
**Rating:** 8

**Review:**

This paper proposes a construct for probabilistic programming languages (termed ‘stat’) that takes as input an initial probability distribution and a Markov kernel, and outputs the unique stationary distribution corresponding to the Markov kernel. The construct augments a previously-proposed PPL (by Staton et al., 2016), and enables a formal description of approximate MCMC compilers for PPLs. While I did not follow every mathematical detail, and am uncertain as to the paper’s novelty, it appears to me to be interesting, well-organized, and rigorous.

---

### Official Review · AnonReviewer2 · 2019-09-30
**New construct in a PPL, enabling error bounds for MCMC samplers**

**Confidence:** 3
**Rating:** 8

**Review:**

The article introduces a PPL with a new primitive, "stat", that is used to explicitly represent the stationary distribution of Markov chains. They prove that probabilistic programs using the "norm" construct can be transformed to use "stat" instead. This makes it possible to represent nested MCMC in a way that enables quantitative analyses of approximation errors, and convergence guarantees.
I find it really interesting to have a representation that enables to quantify (or even estimate) the errors that would be induced by the compiler, even before the program is compiled. This may even better inform the PPL compiler's choices if enough information is provided.
The article is well written, its scope is explicit and well motivated. It exposes the existing and new constructs in a clear way, even for an audience unfamiliar with PPLs, which is something I would like to be kept (maybe pushed further) if selected for an oral.

---

### Decision · Program_Chairs · 2019-10-01

**Decision:**

Accept (Poster)

**Comment:**

The reviewers were impressed with the presentation of this work and argued it was a strong contribution that fits within the scope of the workshop.